# Longitudinal associations between marine omega-3 supplement users and coronary heart disease in a UK population-based cohort

Marleen A H Lentjes,[1] Ruth H Keogh,[2] Ailsa A Welch,[3] Angela A Mulligan,[1] Robert N Luben,[1] Nicholas J Wareham,[4] Kay-Tee Khaw[5]

[1]Department of Public Health and Primary Care, University of Cambridge, UK
[2]Department of Medical Statistics, London School of Hygiene and Tropical Medicine, London, UK
[3]Department of Population Health and Primary Care, University of East Anglia, Norwich, UK
[4]MRC Epidemiology Unit, University of Cambridge, Institute of Metabolic Science, Cambridge, UK
[5]School of Clinical Medicine, University of Cambridge, Addenbrooke's Hospital, Cambridge, UK

**Correspondence to**
Dr Marleen A H Lentjes;
marleen.lentjes@phpc.cam.ac.uk

## ABSTRACT

**Objectives** Assess the association between marine omega-3 polyunsaturated fatty acid (n-3 PUFA) intake from supplements, mainly cod liver oil, and coronary heart disease (CHD) mortality.

**Design** Prospective cohort study, with three exposure measurements over 22 years.

**Setting** Norfolk-based European Prospective Investigation into Cancer (EPIC-Norfolk, UK).

**Participants** 22 035 men and women from the general population, 39–79 years at recruitment.

**Exposure** Supplement use was assessed in three questionnaires (1993–1998; 2002–2004; 2004–2011). Participants were grouped into non-supplement users (NSU), n-3 PUFA supplement users (SU+n3) and non–n-3 PUFA supplement users (SU-n3). Cox regression adjusted for time-point specific variables: age, smoking, prevalent illnesses, body mass index, alcohol consumption, physical activity and season and baseline assessments of sex, social class, education and dietary intake (7-day diet diary).

**Primary and secondary outcome measures** During a median of 19-year follow-up, 1562 CHD deaths were registered for 22 035 included participants.

**Results** Baseline supplement use was not associated with CHD mortality, but baseline food and supplement intake of n-3 PUFA was inversely associated with CHD mortality after adjustment for fish consumption. Using time-varying covariate analysis, significant associations were observed for SU+n3 (HR: 0.74, 95% CI 0.66 to 0.84), but not for SU-n3 versus NSU. In further analyses, the association for SU+n3 persisted in those who did not take other supplements (HR: 0.83, 95% CI 0.71 to 0.97). Those who became SU+n3 over time or were consistent SU+n3 versus consistent NSU had a lower hazard of CHD mortality; no association with CHD was observed in those who stopped using n-3 PUFA-containing supplements.

**Conclusions** Recent use of n-3 PUFA supplements was associated with a lower hazard of CHD mortality in this general population with low fish consumption. Residual confounding cannot be excluded, but the findings observed may be explained by postulated biological mechanisms and the results were specific to SU+n3.

### Strengths and limitations of this study

► The use of dietary supplements was measured three times over the course of 22 years.
► The repeated measures enabled to study change in supplement use in relation to hard endpoints.
► The cohort was population-based as opposed to trials on omega-3 supplement use which have been in high-risk populations.
► Total omega-3 intake from both food (mainly fish sources) and supplement sources could be quantified; however, only for the baseline measure.
► Estimates were adjusted for sociodemographic and other behavioural risk factors, although residual confounding cannot be excluded.

## INTRODUCTION

The traditionally high omega-3 polyunsaturated fatty acid (n-3 PUFA) intake among the Inuit has been associated with a lower risk of ischaemic heart disease mortality in this population.[1] Research leading on from this observed that in a Dutch cohort, where average fish consumption was 20 g/day, higher fish consumption was associated with lower relative risk of coronary heart disease (CHD) mortality between 1960–1980.[2] A recent review of prospective cohort studies has observed a 16% (95% CI 5% to 25%) lower risk of CHD mortality in those consuming one portion of fish per week compared with lower frequency of consumption; however, null findings among included studies were equally observed.[3] The association between fish intake and CHD has been attributed to n-3 PUFA, which are mainly derived from oily fish. According to the National Diet and Nutrition Survey in 2000/2001, less than 50% of the UK population consumed oily fish once a week[4]; however, dietary supplements may provide 50% of the n-3 PUFA intake.[5]

N-3 PUFA at pharmaceutical doses reduce plasma triglycerides and possibly thrombosis; whereas lower doses (around 250 mg/day, obtainable from fish consumption and low dose supplements such as cod liver oil (CLO)) have been observed to lower the risk of sudden death or CHD death in trials and observational studies, which may be due to the influence of n-3 PUFA on heart rate and arrhythmia.[6–8] Reviews of trials on n-3 PUFA supplementation in relation to overall cardiovascular diseases (CVD),[9] or when separated into subgroups of stroke and coronary diseases,[10–12] have shown no overall benefit, particularly in the most recent trials. It has been suggested that background medication, inclusion of non-fatal events as outcome measures and increasing trends in n-3 PUFA/ fish consumption might underlie the null findings.[13 14]

Although there have been a number of studies on the association between n-3 PUFA intake and CHD, the evidence for a protective association between n-3 PUFA supplement intake and CHD remains inconclusive.[15] Consumption of n-3 PUFA supplements is traditionally high in Northern European countries.[16] A Swedish cohort observed no association for fish oil supplements and CVD mortality[17]; whereas a study based in Iceland observed a lower risk of CHD hospitalisation with high frequency of CLO consumption.[18] Previous studies have relied on single baseline measures of supplement use and covariates,[19] excluded fish oil supplement users because of minimal use in the studied cohort[20] or lacked detailed data on supplement use and therefore imputed n-3 PUFA content[21]; moreover, trials have included participants at increased risk of CHD[22–24] or included those who have already experienced a CHD event[25–28]; the effects of n-3 PUFA supplementation in a general population are less well documented, and the first primary prevention trial is ongoing.[29]

The aim of this analysis was to use three repeated measures of supplement use to investigate the association between n-3 PUFA supplement intake and CHD mortality in a general population-based cohort.

## METHODS
An extended version of the methods has been included in online supplementary appendix I.

### Study design
Recruitment for the Norfolk-based European Prospective Investigation into Cancer and Nutrition (EPIC-Norfolk) started in 1993 and was completed in 1998.[30 31] Thirty-five general practices in the Norfolk area of East Anglia (UK) took part in the study; 77 630 registered patients were approached, 30 445 consented. Exposures have been reassessed over the course of the study; participants completed general Health and Lifestyle Questionnaires (HLQ) and attended three health examinations (HE) up to 2011 (figure 1, online supplementary appendix I). Data on health outcomes and mortality were collected passively. Ethical approval for the study was given by the

Norwich District Health Authority Ethics Committee. Participants gave written informed consent.

### Assessment of n-3 PUFA supplement use
Dietary supplements were defined according to the EU directive 2002/46/EC.[32] Prescribed medication containing minerals and/or vitamins were not considered supplements. The sum of eicosapentaenoic acid (EPA) and docosahexaenoic acid (DHA) from supplements were the main exposure, referred to as n-3 PUFA. We used data from three dietary supplement assessments (DSA), named DSA1, DSA2, DSA3. All covered a 1 week recall (online supplementary appendix I).

At each DSA, participants were grouped by the type of supplement they consumed:
► non-supplement users (NSU);
► non-n-3 PUFA supplement users (SU-n3): participants who consumed one or more supplements that did not contain n-3 PUFA;
► n-3 PUFA supplement users (SU+n3): participants who used n-3 PUFA supplements, either singly or in combination with other non-n-3 PUFA supplements.

Participants were also grouped into five categories identifying change in n-3 PUFA supplement use between two consecutive DSA (online supplementary appendix I and appendix II): 'consistent NSU', 'was SU+n3', 'became SU+n3', 'consistent SU+n3', 'other SU'.

Average daily nutrient composition from supplements was calculated using the Vitamin and Mineral Supplement system[33] and added to the average daily food intake to obtain average total nutrient intake. Validation of these data with biomarkers were undertaken.[5]

### Dietary covariates
The analyses adjust for potential dietary confounders measured at DSA1. Diet was assessed using a 7-day diet diary (7dDD) between 1993–1998.[34] Data were entered by trained staff into DINER[35] and checked and calculated by nutritionists using DINERMO.[36] The underlying food composition tables were derived from McCance & Widdowson's Composition of Foods and supplements.[37] Missing values for fatty acids were further completed using a recipe calculation system.[38 39] The baseline 7dDD provided data on average daily energy intake (MJ/day), proportion of energy provided by macronutrients (%en), the sum of EPA and DHA intake (g/day) referred to as 'n-3 PUFA' and consumption of the disaggregated amounts of fruit, vegetable, red meat, processed meat, white meat, and white and oily fish consumption (g/day).[36]

### Assessment of other covariates
Social class was measured through occupational status (professional, managerial, skilled non-manual, skilled manual, semi-skilled, non-skilled). Highest education level achieved was divided in four categories (no qualification, O-level, A-level, degree). Both variables were derived from HLQ1 (coinciding with DSA1) and are time-fixed. The remaining covariates are time-dependent and

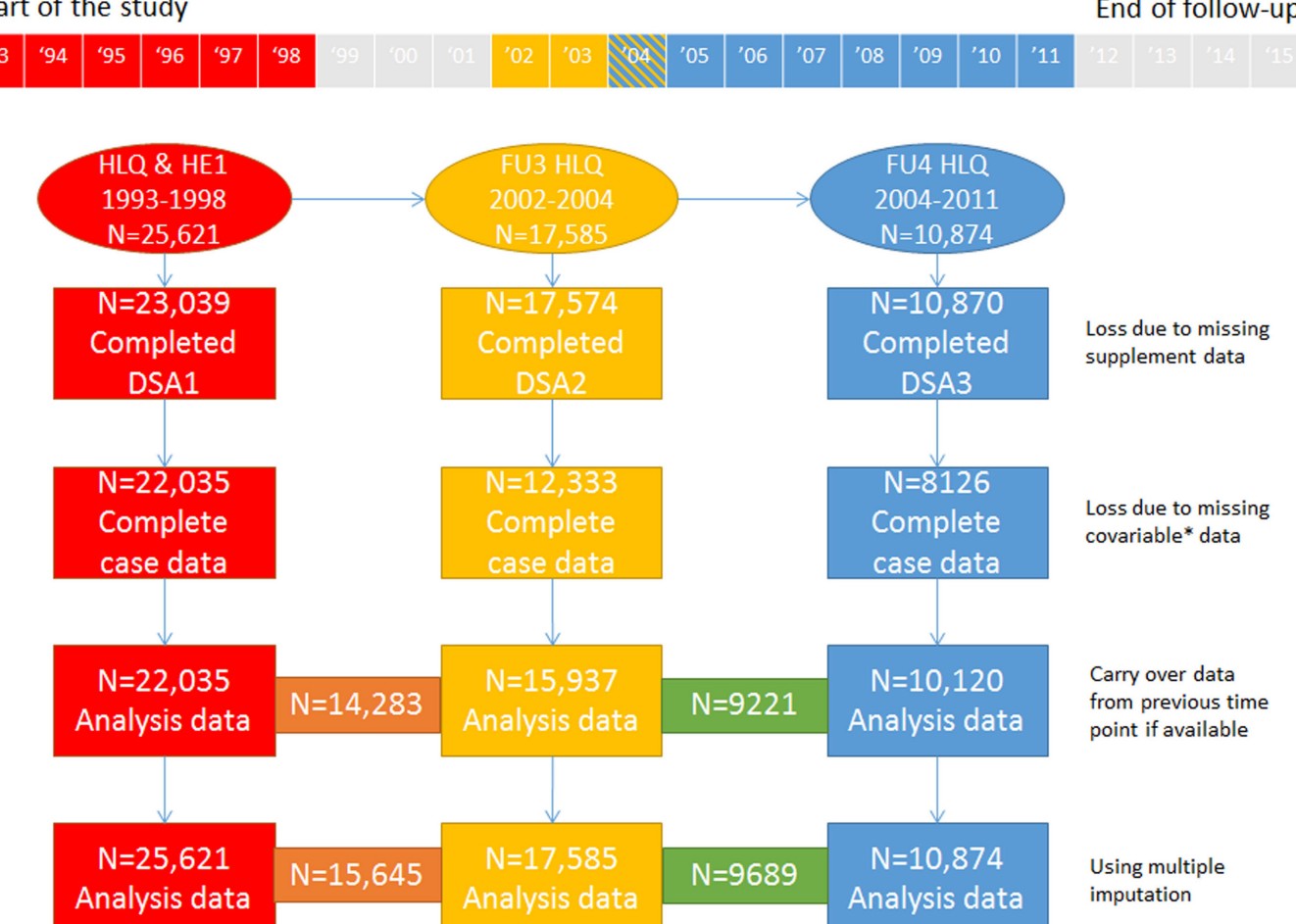

**Figure 1** Number of participants included at each DSA time point studied, with different scenarios of exclusion due to missing data. *Social class, education, smoking status, physical activity, body mass index, marital status and alcohol consumption. The numbers in the orange and green boxes relate to the number of participants available when the two consecutive DSA were considered in the analysis. The grey area in the timeline represents unobserved time. For more information regarding data availability see supplementary appendix I. DSA, dietary supplement assessment; FU, follow-up; HE, health examination; HLQ, Health and Lifestyle Questionnaire.

refer to measures obtained at time points corresponding to the DSA (online supplementary appendix I). We identified participants at higher risk of CHD using responses to the questions, "Has the doctor ever told you that you have [myocardial infarction/diabetes/stroke]?". Participants were classified as a never smoker, former smoker or current smoker. Alcohol consumption was classified as none, >0–14, >14–28, >28 units/week. Marital status was recategorised into married (married/living as married) or non-married (widowed, divorced, separated, single). Participants were classified as being active, moderately active, moderately inactive or inactive. Body mass index (BMI) (kg/m$^2$) was calculated from height and weight measured by a trained nurse.

### Participant selection, case ascertainment and outcomes studied

Participants were eligible for analyses if they provided data on supplement use at any of the three time points (DSA1, DSA2, DSA3) and attended the corresponding HE (figure 1); 4030 participants were not included at any time point studied. The participant's National Health Service number (a unique national patient identifier) was linked to the data from the Office of National Statistics to obtain vital status and causes of death. A similar procedure was followed for causes of hospital admissions registered by the East Norfolk Primary Healthcare Trust which records for its residents the admissions in England and Wales. The main endpoint studied was CHD mortality mentioned anywhere on the death certificate, identified with International Classification of Diseases (ICD) 9 410–414 or ICD10 I20-I25. In sensitivity analyses, we investigated the use of different definitions for the CHD endpoint: (1) CHD mortality mentioned as underlying cause of death, (2) death due to acute myocardial infarction (ICD-codes 410 or I21) or (3) first recorded hospitalisation due to CHD.

### Statistical analysis

A description of the cohort at DSA1, DSA2 and DSA3 is given. To study potential changes in supplement user characteristics over time, a multinomial logistic regression

analysis was performed at each DSA with supplement use group as the outcome variables and the variables noted above as explanatory variables.

The main analyses used Cox regression, using the follow-up time scale. Individuals who died from causes other than the event of interest were censored at their date of death and individuals who did not die from any cause were censored at the end of follow-up (31 March 2015).

First, we modelled associations between supplement use at DSA1 and CHD mortality, with adjustment for covariates measured at DSA1. A series of cumulative adjustment models were used: sex and age-adjusted estimates (model 1); including smoking, BMI $(kg/m^2)$, physical activity, alcohol intake, social class, education and season in which the questionnaire was completed (model 2); including self-reported myocardial infarction, stroke or diabetes (model 3); including energy intake (MJ/day) and disaggregated fruit (g/day), vegetable (g/day), red meat and processed meat (g/day), and white meat (g/day) (model 4); including white fish (g/day) and oily fish (g/day) (model 5). Second, models 1–5 were fitted using quintiles of n-3 PUFA intake from food and supplements at DSA1 as the main exposure (online supplementary appendix III).

The above analyses were repeated using time-updated measures of supplement use and covariates using model 3. First, we performed separate analyses using DSA1, DSA2 and DSA3 as the time origins. Second, we performed time-varying covariates modelling using the full length of follow-up as well as shorter length of follow-up time to acknowledge unobserved changes in individuals' supplement use over time. We equally applied the strategy described here using the change between supplement group categories between DSA.

We assessed interactions between supplement use and sex, age, self-reported illnesses, smoking and BMI using likelihood ratio tests.

At DSA1, participants with missing data on supplement use and covariables were excluded; at DSA2 and DSA3 missing covariates from the last observation were carried forward (figure 1). Alternatively, multiple imputation was used to handle missing data in the Cox regression analyses (supplementary appendix IV). The proportional hazards' assumption was assessed in the time-varying covariates model, by including interactions between time and each explanatory variable. Data were analysed using Stata V.14.

## RESULTS

Analyses were based on 22 035 participants who provided data at DSA1, 15 937 participants at DSA2 and 10 120 participants at DSA3 (figure 1). Median (IQR) of follow-up time was 19 (17, 20) years. During this time, 1562 participants (out of 22 035) died of CHD mentioned anywhere on the death certificate.

## Description of included participants

Table 1 (and online supplementary appendix V) describes the study participants at each DSA. Included participants had a median age of 59, 65 and 69 years at DSA1, DSA2 and DSA3, respectively, and consisted of 45%, 43% and 44% men. We observed a lower smoking prevalence, fewer alcohol consumers and more physical inactivity as time progressed; however, there was still a large range in these behaviours. The proportion of NSU dropped from 61% at DSA1 to 50% at DSA3, with 24%, 31% and 34% using n-3 PUFA containing supplements at the three DSA, respectively. The change in supplement use over time is depicted in online supplementary appendix II.

Multinomial logistic regression analysis assessed the relation of these characteristics to supplement use status, mutually adjusted, at DSA1, DSA2 and DSA3 (figure 2). Having a non-manual social class and a higher education level were associated with SU-n3; whereas older participants, non-smoking, alcohol consumption and fewer self-reported illnesses were asso

ciated with SU+n3. Associations between covariates and supplement use group were similar at the three DSA.

## n-3 PUFA supplement users and hazard of CHD mortality

There was no evidence that the CHD mortality of SU+n3 or SU-n3 differed from NSU when solely using baseline data (online supplementary appendix VI.). Table 2 shows results from the Cox proportional hazards analyses in which the time origin was reset at each DSA. No association was observed between supplement use at DSA1 and CHD mortality for any amount of follow-up time. On the contrary, the HR of CHD mortality for SU+n3 relative to NSU at DSA2 and DSA3 was significantly below 1. With increasing follow-up time the HR attenuated, but remained statistically significant. We observed stronger associations, for all lengths of follow-up studied, between SU+n3 and CHD mortality in the later DSAs.

Using time-varying analysis, we observed a 26% lower hazard among SU+n3 compared with NSU (HR: 0.74, 95% CI 0.66 to 0.84). Significant interactions between follow-up time and age, sex and self-reported myocardial infarction were observed; however, inclusion of these interaction terms resulted in negligible differences in the HR or 95% CI of the main exposure. Results presented are therefore without these interaction terms in the regression equation.

## Change in supplement use over time

Table 3 shows the results from analysis investigating changes in supplement use over time. There was no evidence to suggest that participants who were taking n-3 PUFA supplements at an earlier point in time, but who stopped its use, had a different hazard of CHD mortality compared with consistent NSU (HR$_{DSA1-DSA2}$ 1.24, 95% CI 0.94 to 1.64; HR$_{DSA2-DSA3}$ 1.20, 95% CI 0.79 to 1.81), therefore suggesting that there are no long term benefits of previous n-3 PUFA supplement use. Compared with consistent NSU, participants who reported n-3 PUFA

**Table 1** Characteristics of the EPIC-Norfolk cohort at time of DSA1 (1993–1998), DSA2 (2002–2004) and DSA3 (2004–2011)

| | DSA1 n=22 035 | DSA2 n=15 937 | DSA3 n=10 120 |
|---|---|---|---|
| CHD mortality events from DSA onwards (n) | 1562 | 742 | 241 |
| Total person years at risk (years) | 383 444 | 173 239 | 63 535 |
| Crude cumulative rate (per 1000 person years) | 4.074 | 4.283 | 3.793 |
| Supplement use | | | |
| NSU | 61 (13 444) | 52 (8353) | 50 (5029) |
| SU-n3 | 15 (3263) | 17 (2665) | 16 (1610) |
| SU+n3 | 24 (5328) | 31 (4919) | 34 (3481) |
| n-3 PUFA (g/day)* | | | |
| From food only | 0.12 (0.06–0.34) | – | – |
| From food and supplements combined | 0.16 (0.07–0.40) | – | – |
| Among SU+n3 only | 0.30 (0.17–0.72) | – | – |
| Sex | | | |
| Men | 45 (9890) | 43 (6923) | 44 (4418) |
| Women | 55 (12 145) | 57 (9014) | 56 (5702) |
| Age (years) | 59 (51–67) | 65 (58–72) | 69 (63–76) |
| BMI (kg/m$^2$) | 25.8 (23.7–28.3) | 26.1 (24.0–28.7) | 26.3 (24.0–29.0) |
| Smoking status | | | |
| Current | 11 (2395) | 7 (1149) | 5 (456) |
| Former | 43 (9426) | 50 (7982) | 46 (4671) |
| Never | 46 (10 214) | 43 (6806) | 49 (4993) |
| Social class | | | |
| Professional | 7 (1531) | 8 (1203) | 8 (852) |
| Managerial | 37 (8048) | 39 (6189) | 40 (4047) |
| Skilled non-manual | 17 (3715) | 16 (2585) | 16 (1658) |
| Skilled manual | 23 (5055) | 22 (3483) | 21 (2131) |
| Semi-skilled | 13 (2910) | 13 (2007) | 12 (1163) |
| Non-skilled | 4 (776) | 3 (470) | 3 (269) |
| Marital status | | | |
| Married | 82 (18 127) | 78 (12 394) | 76 (7708) |
| Not married | 18 (3908) | 22 (3543) | 24 (2412) |
| Education level | | | |
| No qualification | 36 (7999) | 32 (5106) | 28 (2851) |
| O-level | 10 (2282) | 11 (1757) | 12 (1184) |
| A-level | 41 (8955) | 42 (6720) | 44 (4427) |
| Degree | 13 (2799) | 15 (2354) | 16 (1658) |
| Season | | | |
| Spring | 26 (5817) | 23 (3645) | 27 (2761) |
| Summer | 25 (5473) | 21 (3295) | 30 (3008) |
| Autumn | 26 (5661) | 32 (5150) | 25 (2513) |
| Winter | 23 (5084) | 24 (3847) | 18 (1838) |
| Physical activity | | | |
| Inactive | 30 (6592) | 36 (5809) | 41 (4106) |
| Moderately inactive | 29 (6389) | 27 (4302) | 28 (2792) |
| Moderately active | 23 (5040) | 20 (3151) | 17 (1709) |

**Table 1** Continued

| | DSA1<br>n=22 035 | DSA2<br>n=15 937 | DSA3<br>n=10 120 |
|---|---|---|---|
| Active | 18 (4014) | 17 (2675) | 15 (1513) |
| Alcohol intake (HLQ) | | | |
| None | 13 (2930) | 32 (5042) | 33 (3312) |
| >0–14 units/week | 72 (15 880) | 52 (8336) | 56 (5704) |
| >14–28 units/week | 11 (2418) | 12 (1950) | 9 (869) |
| >28 units/week | 4 (807) | 4 (609) | 2 (235) |
| Self-reported illness | | | |
| Myocardial infarction | 3 (704) | 5 (717) | 3 (330) |
| Stroke | 1 (304) | 2 (359) | 2 (164) |
| Diabetes | 2 (510) | 5 (745) | 3 (338) |
| Energy—7dDD (MJ/day)* | 8.00 (6.71–9.56) | | |
| Protein (en%) | 15.2 (13.6–16.9) | | |
| Fat (en%) | 33.4 (29.8–36.8) | | |
| Saturated fat (en%) | 12.7 (10.9–14.6) | | |
| Carbohydrates (en%) | 47.6 (43.4–51.5) | | |
| Alcohol (en%) | 2.0 (0–5.9) | | |
| Food intake—7dDD (g/day)* | | | |
| Fruit | 153 (82–242) | | |
| Vegetables | 142 (102–190) | | |
| Red and processed meat | 53 (32–76) | | |
| White meat | 21 (6–37) | | |
| Oily fish | 4 (0–19) | | |
| Consumers only (54%) | 17 (10–30) | | |
| White fish | 13 (0–23) | | |
| Consumers only (66%) | 19 (13–29) | | |
| Total fish | 23 (11–40) | | |
| Consumers only (84%) | 28 (17–44) | | |

Values are % (n) for categorical variables and median (IQR) for continuous variables.

*Data only available from DSA1, see also online supplementary appendix I; for more information on diet at DSA1, see online supplementary appendix V.

7dDD, 7-day diet diary; BMI, body mass index; CHD, coronary heart disease; DSA, dietary supplement assessment; EPIC, European Prospective Investigation into Cancer; HLQ, Health and Lifestyle Questionnaire; n-3 PUFA, omega-3 polyunsaturated fatty acids (sum of eicosapentaenoic acid and docosahexaenoic acid); NSU, non-supplement user; SU, supplement user.

supplement use at both time points, and participants who became a SU+n3 had a lower hazard of CHD mortality. This association was stronger in the time-varying covariates analysis where we observed a HR of 0.69 (95%CI 0.52 to 0.90) among starters of SU+n3 and a HR of 0.78 (95%CI 0.63 to 0.98) among consistent SU+n3. Associations attenuated or became non-significant as follow-up time increased.

### Sensitivity analyses

Using different definitions for the endpoint (figure 3), we observed associations of similar strength for CHD as underlying cause of death (HR: 0.78, 95% CI 0.67 to 0.90) and for acute fatal myocardial infarction when compared with CHD mortality mentioned anywhere on the death certificate. No associations were observed for first event of hospitalisation due to CHD.

Associations between SU+n3 and CHD mortality were similar for men and women, for overweight and non-overweight participants, for participants with and without self-reported myocardial infarction or diabetes and among smokers and non-smokers (online supplementary appendix IV). There was evidence of an interaction between supplement use group and prevalent stroke (p=0.006); although the SU+n3 had similar HRs by prevalent stroke group, SU-n3 had not.

The primary analyses do not distinguish between SU+n3 who do or do not use other supplements. As an additional sensitivity analysis, we investigated this using

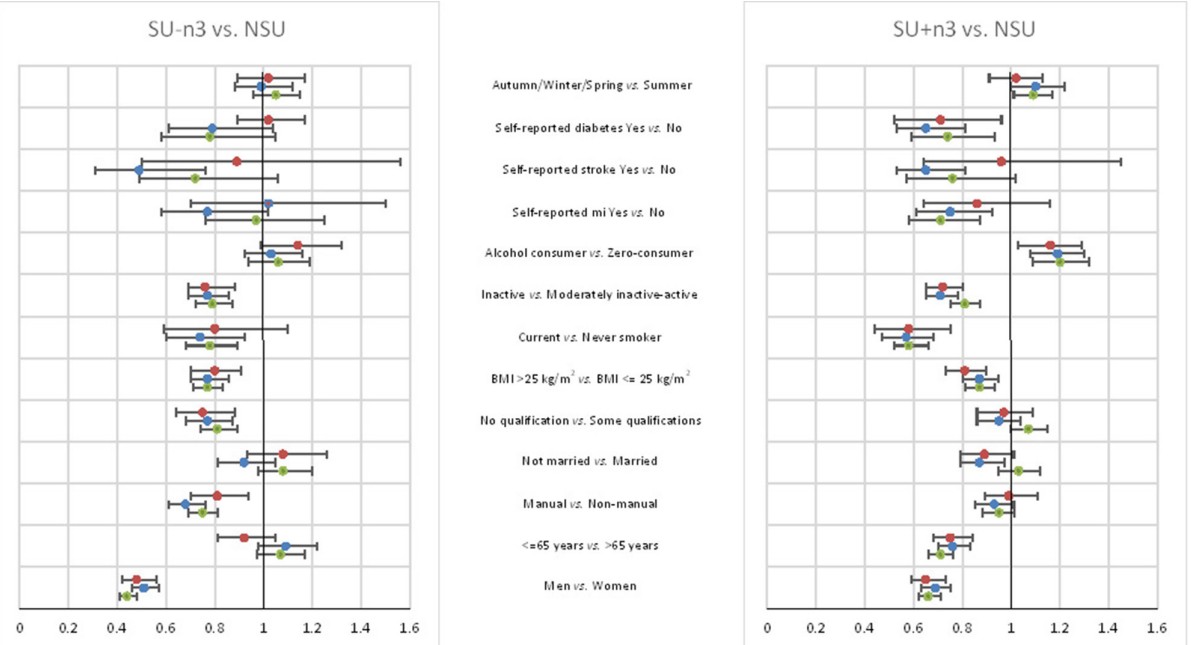

**Figure 2** Characteristics of SU-n3 versus NSU and SU+n3 versus NSU measured at three time points (DSA1 [green], DSA2 [blue], DSA3 [red]) in the EPIC-Norfolk study. Three analyses from multinomial logistic regression at three time points: DSA1 (n=22 035), DSA2 (n=12 333), DSA3 (n=8126). The ORs are mutually adjusted for all variables shown. DSA, dietary supplement assessment; EPIC, European Prospective Investigation into Cancer; NSU, non-supplement user; PUFA, polyunsaturated fatty acids; SU+n3, n-3 PUFA supplement user; SU-n3, supplement user without n-3 PUFA.

a model with binary variables for n-3 PUFA supplement use and non n-3 PUFA supplement use and their interaction, finding that the association between n-3 PUFA supplement use and a reduced hazard persisted in those who used other supplements and those who did not. We found an estimated HR of 0.83 (95% CI 0.71 to 0.97) in those who only use an n-3 PUFA containing supplement versus NSU, a HR of 0.91 (95% CI 0.78 to 1.06) for those who only use non n-3 PUFA containing supplements, and a HR of 0.66 (95% CI 0.56 to 0.79) versus NSU in those who used both, though there was no evidence that the association between n-3 PUFA containing supplements and CHD was different in those taking no other supplements or in those taking other supplements and the HR in those taking n-3 PUFA supplements adjusted for other supplement use was 0.79 (95% CI 0.70 to 0.89).

## DISCUSSION
### Statement of principal findings
We observed a protective association between n-3 PUFA supplement use and CHD mortality in this general population where fish consumption was low and the mean age was approximately 60 years at baseline (DSA1). When data from three DSA were used, lower hazards of CHD mortality were observed for SU+n3 at all three DSA, which were strongest when the follow-up time was short; however, this analysis lacked statistical power. Using a time-varying covariates analysis, we observed a 26% lower hazard of CHD mortality among SU+n3 compared with NSU.

### Strengths and weaknesses of the study
A strength is the selection of participants at baseline from a general population as opposed to fish oil trials which have consisted of high-risk populations.[11] Measures of covariates and supplement use were available from three time points, which enabled mitigation of potential bias due to changes in supplement use over time. We could adjust for several variables representing health behaviours, as well as comorbidities, to account for potential confounding. The DSA had short recall periods of 1 week; however, we have previously shown that measures obtained from three instruments at baseline (DSA1), with recall times varying from 1 year to 1 week, had good agreement.[40] Supplement use in general is associated with health behaviours[41] and heterogeneity across supplement users exists, we therefore separated participants into SU+n3 and SU-n3 and carefully assessed potential confounders at all DSA. A further strength was our use of multiple imputation to handle missing data (online supplementary appendix IV.).

The EPIC-Norfolk study has limitations. The baseline response rate was 39%; however, participants were found to be representative of the Health Survey for England population regarding anthropometry, blood pressure and blood lipids.[30] There was loss to follow-up; however, outcomes were ascertained for those who no longer actively participated. Since the exact dates of supplement use were not known as were the exact dates of self-reported myocardial infarction, stroke or diabetes, we are unable to say which happened first and therefore

**Table 2** The association between supplement use reported at DSA1, DSA2 and DSA3 and subsequent hazard of CHD mortality (where cause of death was mentioned anywhere on the death certificate) adjusted using model 3*

| | Total | Two years of follow-up | | Four years of follow-up | | Follow-up time until next DSA† | | Full follow-up time | |
|---|---|---|---|---|---|---|---|---|---|
| | | CHD Events | | CHD Events | | CHD Events | | CHD Events | |
| | N | N | HR (95% CI) | N | HR (95% CI) | N | HR (95% CI) | N | HR (95% CI) |
| DSA1 (1993–1998) | 22 035 | 71 | | 174 | | 872 | | 1562 | |
| NSU | 13 444 | 47 | 1.00 | 113 | 1.00 | 580 | 1.00 | 1012 | 1.00 |
| SU-n3 | 3263 | 9 | 1.24 (0.60 to 2.56) | 20 | 1.17 (0.73 to 1.90) | 101 | 1.06 (0.85 to 1.31) | 178 | 0.96 (0.82 to 1.13) |
| SU+n3 | 5328 | 15 | 0.95 (0.53 to 1.72) | 41 | 1.05 (0.73 to 1.52) | 191 | 0.86 (0.73 to 1.02) | 372 | 0.94 (0.83 to 1.06) |
| DSA2 (2002–2004) | 15 937 | 75 | | 171 | | 527 | | 742 | |
| NSU | 8353 | 53 | 1.00 | 115 | 1.00 | 346 | 1.00 | 475 | 1.00 |
| SU-n3 | 2665 | 10 | 1.01 (0.51 to 2.02) | 25 | 1.24 (0.79 to 1.93) | 61 | 0.94 (0.71 to 1.24) | 81 | 0.84 (0.66 to 1.07) |
| SU+n3 | 4919 | 12 | 0.55 (0.29 to 1.04) | 31 | 0.64 (0.43 to 0.96) | 120 | 0.75 (0.61 to 0.93) | 186 | 0.81 (0.68 to 0.97) |
| DSA3 (2004–2011) | 10120 | 59 | | 125 | | 241 | | | |
| NSU | 5029 | 44 | 1.00 | 82 | 1.00 | 148 | 1.00 | | |
| SU-n3 | 1610 | 6 | 0.60 (0.25 to 1.42) | 19 | 1.01 (0.61 to 1.67) | 35 | 1.00 (0.69 to 1.45) | | |
| SU+n3 | 3481 | 9 | 0.38 (0.18 to 0.78) | 24 | 0.52 (0.33 to 0.83) | 58 | 0.70 (0.52 to 0.95) | | |
| Time-varying | 24330 | 205 | | 470 | | 1640 | | | |
| NSU | | 144 | 1.00 | 310 | 1.00 | 1074 | 1.00 | | |
| SU-n3 | | 25 | 0.93 (0.61 to 1.43) | 64 | 1.13 (0.86 to 1.49) | 197 | 0.91 (0.78 to 1.06) | | |
| SU+n3 | | 36 | 0.59 (0.41 to 0.85) | 96 | 0.73 (0.58 to 0.92) | 369 | 0.74 (0.66 to 0.84) | | |

The follow-up time in the EPIC-Norfolk study was from 1993 to 2015.

We performed the analysis in two ways. First, in separate analyses using DSA1, DSA2 and DSA3 as the time origins, such that a single DSA was used to predict future CHD mortality (top three sections). Second, we performed an analysis which combines the three DSA into a single analysis. In more detail, we used the most up-to-date exposure and covariate measures for each individual at each time they were at risk, that is, time-varying covariates modelling. In the 'time-varying' approach, the follow-up time is divided by the dates of the respective DSA measures. Each participant is allocated to an exposure group ('NSU', 'SU-n3' or 'SU+n3'), but only for that section of the follow-up time in which they belonged to that group. If they changed supplement use (ie, 'varied' by stopping a supplement or changing the type of supplement) the next section of the follow-up time (until the next DSA) was allocated to the exposure group they changed to. This type of analysis reduces misclassification of the exposure and any other confounders over time in a single analysis. Please note, the variables in this analysis do not explain what the associated risk is when changing from a specific supplement user category to another, this is shown in table 3. The reshaped dataset for time-varying analyses contains a larger number of participants (n=24 330) than available at DSA1 alone since some participants were excluded from DSA1 due to missing covariate data when these covariates were available at DSA2 and/or DSA3 and so the participant contributed follow-up time from DSA2 and/or DSA3 only.

*Using adjustment model 3: time-point specific age, smoking, BMI, alcohol consumption, physical activity, season of questionnaire completion, marital status and self-report of myocardial infarction, stroke or diabetes; as well as: sex, social class and education measured at DSA1.

†In case of DSA3, the censor date was the date of administrative follow-up (31 March 2015).

DSA, Dietary Supplement Assessment; EPIC, European Prospective Investigation into Cancer; NSU, non-supplement users; PUFA, polyunsaturated fatty acids; SU+n3, n-3 PUFA supplement users (mainly cod liver oil); SU-n3, non-n-3 PUFA supplement users.

**Table 3** The association between change in supplement use and subsequent hazard of CHD mortality (where cause of death was mentioned anywhere on the death certificate)

| | Total | Two years of follow-up | | Four years of follow-up | | Follow-up time until next DSA* | | Full follow-up time | |
|---|---|---|---|---|---|---|---|---|---|
| | | Events | | Events | | Events | | Events | |
| | N | N | HR (95% CI) | N | HR (95% CI) | N | HR (95% CI) | N | HR (95% CI) |
| Change between DSA1-DSA2 | 14 283 | 71 | | 162 | | 477 | | 672 | |
| Consistent NSU | 5861 | 42 | 1.00 | 89 | 1.00 | 240 | 1.00 | 335 | 1.00 |
| Were SU+n3 | 1297 | 9 | 1.03 (0.50 to 2.12) | 21 | 1.13 (0.70 to 1.82) | 64 | 1.24 (0.94 to 1.64) | 81 | 1.12 (0.88 to 1.43) |
| Became SU+n3 | 2318 | 6 | 0.58 (0.25 to 1.38) | 18 | 0.79 (0.47 to 1.31) | 46 | 0.69 (0.50 to 0.94) | 75 | 0.77 (0.60 to 1.00) |
| Consistent SU+n3 | 2213 | 6 | 0.51 (0.22 to 1.21) | 13 | 0.50 (0.28 to 0.90) | 65 | 0.87 (0.66 to 1.15) | 96 | 0.88 (0.70 to 1.11) |
| Other type of SU | 2594 | 8 | 0.71 (0.33 to 1.53) | 21 | 0.90 (0.56 to 1.46) | 62 | 0.93 (0.70 to 1.24) | 85 | 0.86 (0.67 to 1.09) |
| Change between DSA2-DSA3 | 9221 | 58 | | 118 | | 218 | | | |
| Consistent NSU | 3427 | 37 | 1.00 | 59 | 1.00 | 107 | 1.00 | | |
| Were SU+n3 | 932 | 5 | 0.58 (0.23 to 1.49) | 17 | 1.22 (0.71 to 2.11) | 29 | 1.20 (0.79 to 1.81) | | |
| Became SU+n3 | 1173 | 2 | 0.22 (0.05 to 0.94) | 8 | 0.54 (0.25 to 1.12) | 17 | 0.63 (0.38 to 1.06) | | |
| Consistent SU+n3 | 2037 | 7 | 0.41 (0.18 to 0.93) | 15 | 0.54 (0.31 to 0.96) | 37 | 0.75 (0.52 to 1.10) | | |
| Other type of SU | 1652 | 7 | 0.57 (0.25 to 1.29) | 19 | 0.97 (0.58 to 1.65) | 28 | 0.79 (0.52 to 1.21) | | |
| Time-varying† | 15 356 | 129 | | 280 | | 695 | | | |
| Consistent NSU | | 79 | 1.00 | 148 | 1.00 | 347 | 1.00 | | |
| Were SU+n3 | | 14 | 0.81 (0.46 to 1.44) | 38 | 1.18 (0.82 to 1.69) | 93 | 1.25 (0.99 to 1.57) | | |
| Became SU+n3 | | 8 | 0.42 (0.20 to 0.88) | 26 | 0.70 (0.46 to 1.07) | 63 | 0.69 (0.52 to 0.90) | | |
| Consistent SU+n3 | | 13 | 0.46 (0.26 to 0.84) | 28 | 0.53 (0.35 to 0.80) | 102 | 0.78 (0.63 to 0.98) | | |
| Other type of SU | | 15 | 0.64 (0.37 to 1.13) | 40 | 0.92 (0.65 to 1.32) | 90 | 0.85 (0.68 to 1.08) | | |

The follow-up time in the EPIC-Norfolk study was from 2002 for DSA1-DSA2 and from 2004 for DSA2-DSA3 (until 2015).

Using adjustment model 3: covariates are from the latest considered time-point (eg, DSA2, when analysing change between DSA1 and DSA2): age, smoking, BMI, alcohol consumption, physical activity, season of questionnaire completion, marital status and self-report of myocardial infarction, stroke or diabetes; as well as: sex, social class and education measured at DSA1.

*In case of DSA3, the censor date was the date of administrative follow-up (31 March 2015).

†The reshaped dataset contains a larger number of participants (N=15 356), since participants did not complete DSA1, but did complete DSA2 and/or DSA3; equally, some participants were excluded from DSA1 due to missing covariate data when these covariates were available at DSA2 and/or DSA3 and so the participant contributed follow-up time from DSA2 and/or DSA3 only. DSA, dietary supplement assessment; EPIC, European Prospective Investigation into Cancer; NSU, non-supplement users; PUFA, polyunsaturated fatty acids; SU+n3, n-3 PUFA supplement users (mainly cod liver oil); SU-n3, non-n-3 PUFA supplement users.

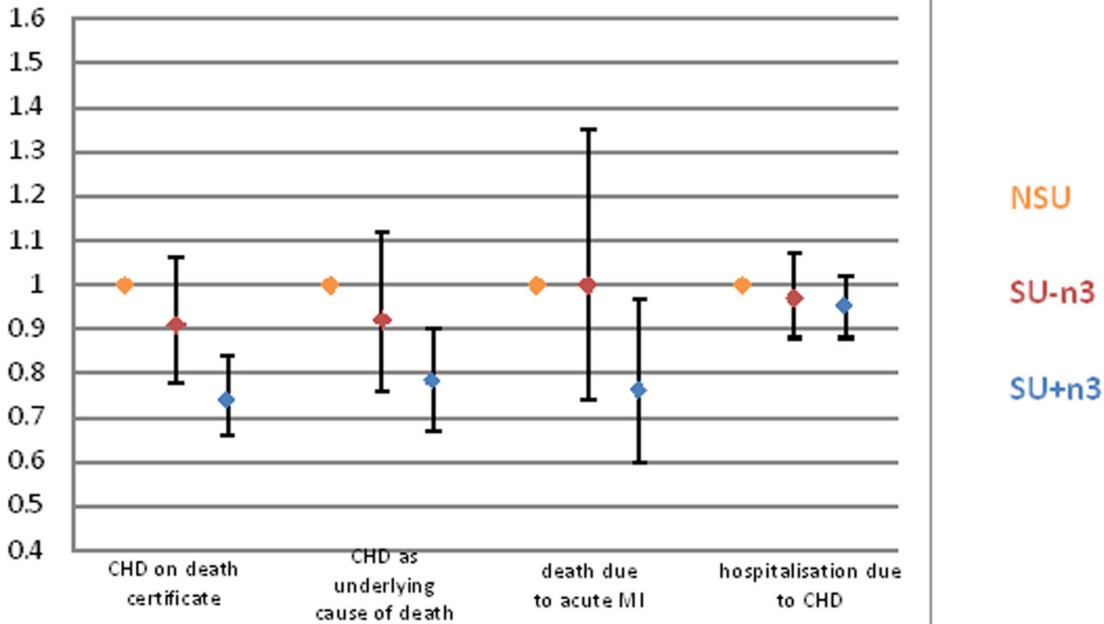

**Figure 3** Time-varying covariate analysis of the association between supplement use and hazard of CHD in the EPIC-Norfolk study (follow-up time from 1993 to 2015). (1) CHD mentioned anywhere on the death certificate (n=1640/24 330, as table 2). (2) CHD mentioned as underlying cause of death on death certificate (n=1084/24 330). (3) Acute myocardial infarction as underlying cause of death on death certificate (n=411/24 330). (4) Hospitalisation due to CHD (n=4087/24 217). The reshaped dataset for mortality analysis contains a larger number of participants (n=24 330) than available at DSA1 alone and therefore more events, since participants did not complete DSA1, but did complete DSA2 and/or DSA3; equally, some participants were excluded from DSA1 due to missing covariate data whereas these covariates were available at DSA2 and/or DSA3 and so the participant contributed follow-up time from DSA2 and/or DSA3 only. The reshaped dataset for hospitalisation analysis contains a smaller number of participants (n=24 216) than the mortality analysis, since participants who did not complete DSA1, but completed DSA2 or DSA3—however had a non-fatal event before DSA2 or DSA3, respectively—were excluded. Using adjustment model 3: time-point specific age, smoking, BMI, alcohol consumption, physical activity, season of questionnaire completion, marital status and self-report of myocardial infarction, stroke or diabetes; as well as: sex, social class and education. NSU, non-supplement users; SU-n3, non-N-3 PUFA supplement users; SU+n3, N-3 PUFA supplement users (mainly cod liver oil); MI, myocardial infarction.

the chronological ordering is unknown. Healthier individuals might have been more likely to continue active participation; however, we adjusted for a number of factors to mitigate potential bias arising due to this. The time intervals between DSA were relatively long, particularly between DSA1 and DSA2; therefore, misclassification of the exposure is likely to have occurred. Data on fish consumption (and hence n-3 PUFA intake from food) from 7dDD could only be estimated at baseline (online supplementary appendix III). We considered the observed association between the SU+n3 and CHD mortality unlikely to be due to the nutrient/compound content of other non-n-3 PUFA supplements being used, and indeed in sensitivity analyses we found a significant association between n-3 PUFA supplement use and a reduced hazard in those using no other supplements. We also considered it unlikely that the observed associations could be due to the additional vitamins found in n-3 PUFA containing supplements, since these typically contain a 100% of the recommended nutrient intake and such doses have not been associated with cardiovascular mortality.[42 43] Although we found a protective association between SU-n3 and a reduced hazard for CHD mortality

in sensitivity analysis (0.87, 95% CI 0.76 to 0.98), the evidence for this was weak and we were unable to study the wide variety of individual nutrients/compounds in these supplements and we cannot rule out from these results that other supplements could be associated with CHD mortality. Identification of a specific nutrient and biological pathway, as well as the potential for a relative change in the type of supplement consumed among the SU-n3, requires additional analysis with emphasis on confounding by lifestyle and other factors, possibly particularly so if more than one non-n-3 PUFA supplement was consumed. Residual confounding due to dietary differences between SU+n3, SU-n3 and NSU, particularly those occurring over time, such as differences in fruit, vegetables and meat, also cannot be excluded. Lastly, the analyses required interpretation of many models, which will have increased the role of chance findings; however, the significant findings observed may be explained by biological mechanisms, the results were specific to SU+n3 and we observed significant associations of starting or continued use of n-3 PUFA supplements with a lower hazard for CHD mortality.

## Meaning of the study: possible explanations and implications for clinicians and policy-makers

The associations observed at the low dose of supplemental EPA and DHA in this cohort makes an antiarrhythmic function of n-3 PUFA, a likely biological pathway.[6 7] In time-varying covariate analysis, it was observed that n-3 PUFA supplement use was associated with 26% lower hazard of CHD mortality compared with non-supplement use. A similar association has been estimated from trial and prospective cohort data[6] which concluded that 250–500 mg/day of n-3 PUFA lowered risk by 25% or more; a dose which corresponds to the intake from food and supplements of 300 mg/day observed in this cohort among SU+n3 (table 1). We have previously reported a non-significant association between plasma n-3 PUFA and CHD incidence in a sub-cohort of the EPIC-Norfolk study[44]; however, plasma concentrations were only available at baseline and are likely to have fluctuated over time, partly considering the variability in supplement use (online supplementary appendix II). The results between the studies are consistent when only relying on supplement data from DSA1 in the current analysis (online supplementary appendix VI). The VITAL cohort study found no significant association between increasing quartiles of n-3 PUFA intake from food and supplements and CHD ($P_{trend}$=0.120), although a significant trend was observed for those participants free of baseline CHD ($P_{trend}$=0.029). When including all DSA, effect modification for prevalent myocardial infarction was not observed in EPIC-Norfolk, which might be indicative of the different motivations for CLO use in the UK versus fish oil supplement use in the USA.[45 46] In Europe, the highest oily fish consumption is observed in Spain and Scandinavian countries, and is approximately 2–3 fold higher than fish intake in the UK.[47] A Norwegian cohort, where 30%–41% consumed two or more fish meals per week, observed no association between CLO consumption and CHD mortality.[19] Fish could therefore have exceeded the threshold for an antiarrhythmic effect in this Norwegian cohort to which CLO could not add, but might supply a considerable contribution to n-3 PUFA intake in the UK. The AGES study in Iceland, where the majority of women consumed 2–4 fish portion per week (85% lean fish), did observe a lower risk of hospitalisation due to CHD when n-3 PUFA from CLO was ≥5.9 g/week, whereas no association was observed with fish consumption.[18] We did not observe any association with CHD hospitalisation; however, the n-3 PUFA dose in the AGES study was much higher and therefore efficacy might have been through atherosclerotic pathways. Fish oil supplements cannot contain all the nutrients that would be obtained when consuming fish, or indeed the other meal components consumed with it (eg, vegetables) or the red meat replacement that took place,[48 49] but n-3 PUFA supplements contain lower concentrations of contamination,[50] and provide essential fatty acids without any coating and/or frying oils which might supply trans-fatty acids which are positively

associated with CHD.[11] This is consistent with findings that broiled or baked fish have been associated with a lower risk of CHD mortality, whereas fried fish has not.[51] Moreover, from DSA1, it is known that n-3 PUFA supplements were consumed on a daily basis by approximately 95% of the participants.[40] Fish consumption has not been a daily dietary habit in the UK based on National Survey data in 2000/2001 and 2008/2009–2011/2012.[4 52] Incorporation of n-3 PUFA into plasma, platelets and red blood cells has been shown to be enhanced with a daily versus two times a week regimen of similar weekly supplement doses.[53]

## Unanswered questions and future research

Food composition tables which include contaminants (online supplementary appendix III), nutrient composition from supplements and foods and which distinguish between fish preparation methods, could clarify any association between n-3 PUFA sources and CHD to further inform public health dietary guidelines. The results from the first primary prevention trial on n-3 PUFA supplement use are to be expected.[29]

## CONCLUSION

Recent use of n-3 PUFA supplements was associated with a lower hazard of CHD mortality in a population-based cohort with low fish consumption. This association was specific to SU+n3 and independent of history of myocardial infarction.

**Acknowledgements** The authors thank the EPIC-Norfolk participants and the research and administrative staff for their contributions, past and present.

**Contributors** MAHL: wrote the manuscript, prepared and analysed both the supplement and dietary data. Developed the research question. RHK: advised on statistical analysis, specifically time-varying covariates analysis and multiple imputation. AAW: Managed the dietary data collection of the study. Contributed to the study protocol. AAM: Prepared dietary data. RNL: Manages study data. Contributed to the study protocol. NJW: Principal investigator of the study. K-TK: Principal investigator of the study. All authors critically read and contributed to the manuscript.

**Funding** All authors report programme grants from Cancer Research UK (G0401527, G1000143) and the Medical Research Council (MRC) (C864/A8257, C864/A14136) during the study. RHK is supported by an MRC fellowship (MR/M014827/1). The funders of EPIC-Norfolk had no role in the study design, data collection, data analysis, interpretation of data, writing of the report or in the decision to submit the paper for publication.

**Competing interests** None declared.

**Ethics approval** Norwich District Health Authority Ethics Committee.

**Provenance and peer review** Not commissioned; externally peer reviewed.

**Data sharing statement** More information on the EPIC-Norfolk study can be obtained from the website: www.epic-norfolk.org.uk. Requests for collaboration will be discussed at the EPIC-Norfolk management meetings.

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
