## [Reviewer comments · BMJ Open]

ARTICLE DETAILS

TITLE (PROVISIONAL)	Longitudinal associations between marine omega-3 supplement users and Coronary Heart Disease in a UK population-based cohort
AUTHORS	Lentjes, Marleen; Keogh, Ruth; Welch, A. A.; Mulligan, Angela; Luben, Robert; Wareham, Nicholas; Khaw, Kay-Tee

VERSION 1 - REVIEW

REVIEWER	Roberto Latini IRCCS - Istituto di Ricerche Farmacologiche Mario Negri Milano Italia
REVIEW RETURNED	12-May-2017

GENERAL COMMENTS	present the results of a community based longitudinal study assessing the impact of n-3 PUFA as supplements and diet on cardiovascular adverse events and death. The ms is well written, data and results well presented although the results and tables are not easy to understand, without a master in statistics. The topic is not novel, many studies of all kinds have been conducted in this field since more than 30 years, but still the issue of a benefit from fish oil is controversial. The Authors should make more evident what their study adds to the existing studies, in terms of evidence pro- or con-n-3PUFA. The Authors in the Conclusion make a plea for new studies in the area: the reviewer wonders (and possibly more than one reader) when, if ever, the n-3PUFA-CV health saga will come to an end. The Authors may want to comment on it in their ms. No specific comments.
--

REVIEWER	Joshua Lewis University of Sydney
REVIEW RETURNED	15-May-2017

GENERAL COMMENTS	. The manuscript by Lentjes and colleagues entitled "Longitudinal associations between n-3 containing PUFA supplements and Coronary Heart Disease in a UK population-based cohort" reports the association between PUFA supplement use assessed at three time points with CHD mortality. The manuscript is of interest and well written. Minor a) In the abstract the final sentence reads "Residual confounding cannot be excluded, but associations observed were consistent with the postulated antiarrhythmic effect of low dose n-3 PUFA." However
--

	this manuscript did not investigate antiarrhythmic effects and this sentence should be removed or softened. b) The lack of dietary data at DSA2 and DSA3 and the potential for missing important changes to the diet over such a long follow is mentioned in the limitations but should expanded upon. c) There was no STROBE checklist. d) The authors should be consistent and use either coronary heart disease or ischemic heart disease. e) Within the SU+n-3 users was there power to compare the people that used SU+n-3 singly vs in combination with other non n-3 PUFA supplements? f) Page 13- line 294 brackets should be closed.
--	---

VERSION 1 – AUTHOR RESPONSE

Reviewer: 1

Reviewer Name: Roberto Latini

Institution and Country: IRCCS - Istituto di Ricerche Farmacologiche Mario Negri, Milasno, Italia

Please state any competing interests: None declared

Authors of ms bmjopen-2017-017471 present the results of a community based longitudinal study assessing the impact of n-3 PUFA as supplements and diet on cardiovascular adverse events and death.

The ms is well written, data and results well presented although the results and tables are not easy to understand, without a master in statistics.

We thank the reviewer for this overall positive evaluation. To help future readers, we explained some of the background in the two different statistical approaches in the footnote of Table 2.

The topic is not novel, many studies of all kinds have been conducted in this field since more than 30 years, but still the issue of a benefit from fish oil is controversial. The Authors should make more evident what their study adds to the existing studies, in terms of evidence pro- or con-n-3PUFA. Our results are novel with respect to the primary prevention, longitudinal and observational nature of the study, where the population at baseline was recruited from the general population (wth low fish consumption). This is different from the trials on omega 3 supplements which have only included high risk and secondary prevention populations. These aspects of our study have been highlighted in the 'article summary' section (page 7) and in lines 142-154 of the introduction where we indicate some of the issues in the current publications. In the discussion, we consider the proposed n 3 PUFA threshold of 250 mg/d for an antiarrhythmic effect in relation to fish and supplement intake in the EPIC-Norfolk cohort (lines 388-395). We note the potential pros and cons (and mechanisms) by which n 3 PUFA from fish might defer from supplement sources by describing research from others in the area of daily vs. non-daily supplement use, the potential for contaminants in fish (line 420 and Appendix V) and the preferred fish preparation methods in the UK (lines 420-425). Additional to the CHD-related findings, we addressed a methodological issue. We have highlighted the necessity for repeated measures of dietary supplement use, since relying on a single measure is likely to misclassify participants over time and therefore to attenuate any association.

The Authors in the Conclusion make a plea for new studies in the area: the reviewer wonders (and possibly more than one reader) when, if ever, the n-3PUFA-CV health saga will come to an end. The Authors may want to comment on it in their ms.

We have now re-stated that the results from the first primary prevention trial on omega 3 PUFA still need to be published.

To comply with the word count, a section of the manuscript relating to total (food + supplement) intake of n 3 PUFA, was moved to Appendix V (page 51). Some of the study limitations relating to the single

measure of dietary intake were therefore moved to this Appendix. The conclusion referred to some of the results from Appendix V. This should have been adapted, we propose to remove “The masking of the association between n 3 PUFA and CHD mortality by fish consumption at baseline, through either preparation methods or contaminants, might play a role in the association between n 3 PUFA and CHD mortality, which will require further investigation with time-updated measures.”

No specific comments.

Reviewer: 2

Reviewer Name: Joshua Lewis

Institution and Country: University of Sydney

Please state any competing interests: None declared

Please leave your comments for the authors below

The manuscript by Lentjes and colleagues entitled “Longitudinal associations between n-3 containing PUFA supplements and Coronary Heart Disease in a UK population-based cohort” reports the association between PUFA supplement use assessed at three time points with CHD mortality. The manuscript is of interest and well written.

We thank the reviewer for these positive comments.

Minor

a) In the abstract the final sentence reads “Residual confounding cannot be excluded, but associations observed were consistent with the postulated antiarrhythmic effect of low dose n-3 PUFA.” However this manuscript did not investigate antiarrhythmic effects and this sentence should be removed or softened.

We appreciate the reviewer’s concern. We have changed this sentence to: “Residual confounding cannot be excluded, but the findings observed may be explained by postulated biological mechanisms and the results were specific to SU+n3.”

b) The lack of dietary data at DSA2 and DSA3 and the potential for missing important changes to the diet over such a long follow is mentioned in the limitations but should be expanded upon.

To comply with the word count, a section of the manuscript relating to total (food + supplement) intake of n 3 PUFA, was moved to Appendix V (page 51). Some of the study limitations relating to the single measure of dietary intake were therefore moved to this Appendix and include: lack of data on within person change of fish consumption in the current study, comparison with National Diet and Nutrition Survey, discussion of the results in relation to confounding by indication and contamination of fish. The conclusion refers to some of the results from Appendix V. This should have been adapted, we propose to remove “ The masking of the association between n 3 PUFA and CHD mortality by fish consumption at baseline, through either preparation methods or contaminants, might play a role in the association between n 3 PUFA and CHD mortality, which will require further investigation with time-updated measures.”

c) There was no STROBE checklist.

We apologise, the STROBE list we had uploaded was out of date. This has been corrected.

d) The authors should be consistent and use either coronary heart disease or ischemic heart disease. We chose to use the terminology as described in the paper (there is only one occurrence, we have therefore now removed the abbreviation IHD).

e) Within the SU+n-3 users was there power to compare the people that used SU+n-3 singly vs in combination with other non n-3 PUFA supplements?

Inclusion of single vs. multiple supplement use in baseline models did not change the associations observed. Moreover, at the time of writing, this variable was unavailable beyond baseline. To be consistent with all other supplement use measures presented, variables at all time points have now been cleaned and created. They represent the number of additional non n 3 PUFA supplements consumed. The results of this analysis can be found in the last section of the sensitivity analysis (lines 318-327) and discussion (lines 364-378).

f) Page 13- line 294 brackets should be closed.

This issue does not appear in the Word document, it seems to be created when converting the document to a pdf, which is something that can hopefully be amended in the proofs.

VERSION 2 – REVIEW

REVIEWER	Roberto Latini IRCCS - Istituto Mario Negri, Milano, Italy
REVIEW RETURNED	28-Jun-2017

GENERAL COMMENTS	The Authors have satisfactorily answered to reviewer's questions and addressed the issues raised adequately in the text.
--

REVIEWER	Joshua Lewis University of Sydney, Australia
REVIEW RETURNED	22-Jun-2017

GENERAL COMMENTS	The authors have addressed all my concerns adequately.
--